# Training Neural Networks is $\exists\mathbb{R}$-complete

**Mikkel Abrahamsen**
University of Copenhagen
miab@di.ku.dk

**Linda Kleist**
Technische Universtität Braunschweig
kleist@ibr.cs.tu-bs.de

**Tillmann Miltzow**
Utrecht University
t.miltzow@uu.nl

## Abstract

Given a neural network, training data, and a threshold, finding weights for the neural network such that the total error is below the threshold is known to be NP-hard. We determine the algorithmic complexity of this fundamental problem precisely, by showing that it is $\exists\mathbb{R}$-complete. This means that the problem is equivalent, up to polynomial time reductions, to deciding whether a system of polynomial equations and inequalities with integer coefficients and real unknowns has a solution. If, as widely expected, $\exists\mathbb{R}$ is strictly larger than NP, our work implies that the problem of training neural networks is not even in NP.

Neural networks are usually trained using some variation of backpropagation. The result of this paper offers an explanation why techniques commonly used to solve big instances of NP-complete problems seem not to be of use for this task. Examples of such techniques are SAT solvers, IP solvers, local search, dynamic programming, to name a few general ones.

## 1 Introduction

Training neural networks is a fundamental problem in machine learning. An *(artificial) neural network* is a brain-inspired computing system. Neural networks are modelled by directed acyclic graphs where the vertices are called *neurons*; for an example consider Figure 1.

In this paper, we show that it is $\exists\mathbb{R}$-complete to decide if there exists weights and biases that will result in a cost below a given threshold. Loosely speaking, the complexity class $\exists\mathbb{R}$ is defined as problems that are equivalent to deciding whether a system of polynomial equations and inequalities and many real unknowns has a solution, and it is known that NP $\subseteq \exists\mathbb{R} \subseteq$ PSPACE, although none of these inclusions are known to be strict.

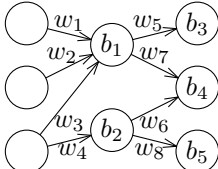 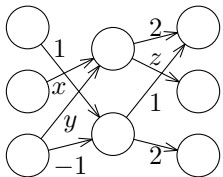

Figure 1: (left) The architecture of a neural network with one layer of hidden neurons. As an example of a training problem, we are given the data points $D = \{(1, 2, 3; 1, 2, 3), (3, 2, 1; 2, 4, 6)\}$, the identity as the activation function $\varphi$ for every neuron, the threshold $\delta = 10$, and the mean of squared errors as cost function. Are there weights and biases such that the total cost is below $\delta$? (right) The architecture of an example instance of RESTRICTED TRAINING. Further, the input consists of the data points $d_1 = (0, 1, 1; 1, 1, ?)$ and $d_2 = (2, 1, 0; ?, ?, 0)$, the activation function $\varphi(x) = x$, the threshold $\delta = 7$, and the Manhattan norm ($\|\cdot\|_1$) as the cost function. If we set the weights to $(x, y, z) = (1, 0, -1)$, we easily compute a total error of $4 + 2 = 6$, which is below the threshold of 7. Thus $(x, y, z) = (1, 0, -1)$ is a valid solution.

35th Conference on Neural Information Processing Systems (NeurIPS 2021).

It is already well known that training neural networks is NP-hard, so one might wonder what we learn from the $\exists\mathbb{R}$-completeness. There exists a large variety of tools for approaching NP-complete problems: SAT solvers, IP solvers, local search, dynamic programming, divide and conquer, prune and search, tree-width based algorithms, or even brute force approaches. Note that local search refers to a technique where the solution is locally changed combinatorially in a small number of components and thus, local search is not related to gradient descent.

In contrast, all research (we are aware of) that actively tries to train neural networks uses variants of backpropagation. Many of the fast and reliable off-the-shelve algorithms solving large instances of NP-complete problems optimally are easy to use, fast, need little to no fine-tuning, and are able to give negative certificates. This makes them much better than backpropagation. So one wonders why backpropagation works better for training neural networks than these strong techniques?

Our result offers the following answers. Assuming NP $\neq \exists\mathbb{R}$ as widely believed, $\exists\mathbb{R}$-hardness gives a strong argument that techniques working for NP-complete problems will not work for training neural networks. As a consequence, our work offers explanations for the observed good experiences with backpropagation when compared to the classical methods.

Neural networks have shown to be superior for solving very complicated tasks, for instance in the context of natural language processing or image recognition. In order to model languages or recognize images, discrete structures (e.g., structures avoiding the use of floating points) are much easier to understand and manipulate and thus in principle preferable over neural networks. This raises the question why we do not use discrete structures like formal grammars in order to describe languages rather than neural networks. So far, we can only reason as follows: Modelling languages using grammars was tried over decades but neural networks outperform formal grammars on all benchmark tasks. Yet, it leaves the question lingering: Have we maybe just not tried hard enough to model image processing and other problems by discrete structures? As mentioned before, it is widely believed that $\exists\mathbb{R}$-complete problems are strictly more difficult than NP-complete problems. In particular, $\exists\mathbb{R}$-complete problems are more expressible and go beyond discrete structures, as they have an inherently continuous nature. In this sense, the fact that training neural networks is complete for $\exists\mathbb{R}$ indicates that neural networks are a more powerful tool than any purely discrete structures, as they are able to model much richer objects.

**Limitations.** While $\exists\mathbb{R}$-completeness gives an interesting perspective on training neural networks, we should also keep the followings facts in mind when interpreting.

- In order to prove hardness, we carefully choose the architecture of our neural network to get an instance that is particularly difficult to train. In practice, one would often use a fully connected neural network. It remains an interesting question if this is also $\exists\mathbb{R}$-complete (see also Section 4).

- In practice, one might not care about finding the exact optimum when training a neural network. For instance, we could ask for a solution that gives a training error of at most OPT $+ \varepsilon$. Here, OPT is the minimum training error and $\varepsilon > 0$ is a small input parameter. We do not know if this version is also hard.

- We show that the problem of exact cost minimization is $\exists\mathbb{R}$-complete. This presupposes that we do all calculations exactly when evaluating the network on an input vector. In practice floating point numbers of $32$ bits, or even as little as $8$ bits [39], are used, which naturally limits the precision of all calculations. Hence, our result does strictly speaking not imply hardness of the problem from a practical perspective.

Given these limitations, we think that it is appropriate to regard $\exists\mathbb{R}$-completeness of the training neural network problem as an *indication* of its practical difficulty, but not as an ultimate truth. Similar limitations hold for many other algorithmic complexity results.

## 1.1 Definition of the training problem

The source nodes of the graph representing a neural network are called the *input neurons* and the sinks are called *output neurons*, and all other neurons are said to be *hidden*. A network computes in the following way: Each input neuron $s$ receives an input signal (a real number) which is sent through all out-going edges to the neurons that $s$ points to. A non-input neuron $v$ receives signals through the

incoming edges, and $v$ then processes the signals and transmits a single output signal to all neurons that $v$ points to. The values computed by the output neurons are the result of the computation of the network. A neuron $v$ evaluates the input signals by a so-called *activation function* $\varphi_v$. Each edge has a *weight* that scales the signal transmitted through the edge. Similarly, each neuron $v$ often has a *bias* $b_v$ that is added to the input signals. Denoting the unweighted input values to a neuron $v$ by $\mathbf{x} \in \mathbb{R}^k$ and the corresponding edge weights by $\mathbf{w} \in \mathbb{R}^k$, then the output of $v$ is given by $\varphi_v(\langle \mathbf{w}, \mathbf{x} \rangle + b_v)$.

During a training process, the network is fed with input values for which the true output values are known. The task is then to adjust the weights and the biases so that the network produces outputs that are close to the ground truth specified by the training data. We formalize this problem in the following definition.

**Definition 1** (Training Neural Networks)**.** *The problem of training a neural network (*NN-Training*) has the following inputs explained above:*

- *A neural network architecture $N = (V, E)$, where some $S \subset V$ are input neurons and have in-degree $0$ and some $T \subset V$ are output neurons and have out-degree $0$,*
- *an activation function $\varphi_v : \mathbb{R} \longrightarrow \mathbb{R}$ for each neuron $v \in V \setminus S$,*
- *a cost function $c : \mathbb{R}^{2|T|} \longrightarrow \mathbb{R}_{\geq 0}$,*
- *a threshold $\delta \geq 0$, and*
- *a set of data points $D \subset \mathbb{R}^{|S|+|T|}$.*

*Here, each data point $\mathbf{d} \in D$ has the form $\mathbf{d} = (x_1, \ldots, x_{|S|}; y_1, \ldots, y_{|T|})$, where $\mathbf{x}(\mathbf{d}) = (x_1, \ldots, x_{|S|})$ specifies the values to the input neurons and $\mathbf{y}(\mathbf{d}) = (y_1, \ldots, y_{|T|})$ are the associated ground truth output values. If the actual values computed by the network are $\mathbf{y}'(\mathbf{d}) = (y'_1, \ldots, y'_{|T|})$, then the cost is $c(\mathbf{y}(\mathbf{d}), \mathbf{y}'(\mathbf{d}))$. The total cost is then*

$$C(D) = \sum_{\mathbf{d} \in D} c(\mathbf{y}(\mathbf{d}), \mathbf{y}'(\mathbf{d})).$$

*We seek to answer the following question. Do there exist weights and biases of $N$ such that $C(D) \leq \delta$?*

We say that a cost function $c$ is *honest* if it satisfies that $c(\mathbf{y}(\mathbf{d}), \mathbf{y}'(\mathbf{d})) = 0$ if and only if $\mathbf{y}(\mathbf{d}) = \mathbf{y}'(\mathbf{d})$. An example of an honest cost function is the popular *mean of squared errors*:

$$c(\mathbf{y}(\mathbf{d}), \mathbf{y}'(\mathbf{d})) = \frac{1}{|T|} \sum_{i=1}^{|T|} (y_i - y'_i)^2.$$

## 1.2 The Result

As our main result, we determine the algorithmic complexity of the fundamental problem NN-Training.

**Theorem 2.** NN-Training *is $\exists \mathbb{R}$-complete, even if*

- *the neural network has only one layer of hidden neurons and three output neurons,*
- *all neurons use the identity function $\varphi(x) = x$ as activation function,*
- *any honest cost function $c$ is used,*
- *each data point is in $\{0, 1\}^{|S|+|T|}$,*
- *the threshold $\delta$ is $0$, and*
- *there are only three output neurons.*

**Organization** We show that NN-Training is contained in $\exists \mathbb{R}$ in Section 2 and prove its hardness in Section 3. We also discusses how our proof could be modified to work with the ReLu activation function. Section 4 contains a discussion and open problems. In the remainder of the introduction, we familiarize the reader with the complexity class $\exists \mathbb{R}$, discuss the *practical* implications of our result, and give an overview of related complexity results on neural network training.

## 1.3 The Existential Theory of the Reals

In the problem ETR, we are given a logical expression (using $\wedge$ and $\vee$) involving polynomial equalities and inequalities with integer coefficients, and the task is to decide if there exist real variables that satisfy the expression. One example is

$$\exists x, y \in \mathbb{R} : (x^2 + y^2 - 6x - 4y + 13) \cdot (y^4 - 4y^3 + x^2 + 6y^2 + 2x - 4y + 2) = 0 \wedge (x \geq 4 \vee y > 1).$$

This is a `yes` instance because $(x, y) = (3, 2)$ is a solution. Due to deep connections to many related fields, ETR is a fundamental and well-studied problem in Mathematics and Computer Science. Despite its long history, we still lack algorithms that can solve ETR efficiently in theory and practice. The *Existential Theory of the Reals*, denoted by $\exists \mathbb{R}$, is the complexity class of all decision problems that are equivalent under polynomial time many-one reductions to ETR. Its importance is reflected by the fact that many natural problems are $\exists \mathbb{R}$-complete. Famous examples from discrete geometry are the recognition of geometric structures, such as unit disk graphs [29], segment intersection graphs [28], stretchability [32, 37], and order type realizability [28]. Other $\exists \mathbb{R}$-complete problems are related to graph drawing [27], statistical inference [9], Nash-Equilibria [20, 7], Geometric-embeddability of simplicial complexes [4] geometric packing [5], the art gallery problem [3], non-negative matrix factorization [36], continuous constraint satisfaction problems [31] and geometric linkage constructions [1]. We refer the reader to the lecture notes by [28] and surveys by [35] and [13] for more information on the complexity class $\exists \mathbb{R}$.

As mentioned before, establishing $\exists \mathbb{R}$-completeness of a problem gives us a better understanding of the inherent difficulty of finding exact algorithms for it. Problems that are $\exists \mathbb{R}$-complete often require irrational numbers of arbitrarily high algebraic degree or doubly exponential precision to describe valid solutions. These phenomena make it hard to find efficient algorithms. We know that NP $\subseteq \exists \mathbb{R} \subseteq$ PSPACE [12], and both inclusions are believed to be strict, although this remains an outstanding open question in the field of complexity theory. In a classical view of complexity, we distinguish between problems that we can solve in polynomial time and *intractable* problems that may not be solvable in polynomial time. Usually, knowing that a problem is NP-hard is argument enough to convince us that we cannot solve the problem efficiently. Yet, assuming NP $\neq \exists \mathbb{R}$, $\exists \mathbb{R}$-hardness proves the limitations of NP methods. To give a simple example, NP-complete problems can be solved in a brute-force fashion by exhaustively going through all possible solutions. Although this method is not very sophisticated, it is good enough to solve small sized instances. The same is not possible for $\exists \mathbb{R}$-complete problems due to their continuous nature. As a second example, the difficulty of solving $\exists \mathbb{R}$-complete problems is nicely illustrated by the problem of placing eleven unit squares into a minimum sized square container without overlap. Whether a given square can contain eleven unit squares can be expressed as an ETR-formula of modest size, so if such formulas could be solved efficiently, we would know the answer (at least to within any desired accuracy). However, it is only known that the sidelength is between $2 + 4/\sqrt{5} \approx 3.788$ and $3.878$ [21].

## 1.4 Previous Hardness Results for NN-Training

It has been known for more than three decades that it is NP-hard to train various types of neural networks for binary classification [8, 30, 25], which means that the output neurons use activation functions that map to $\{0, 1\}$. The first hardness result for networks using continuous activation functions appears to be by Jones [24], who showed NP-hardness of training networks with two hidden neurons using sigmoidal activation functions and one output neuron using the identity function. Hush [23] and Šíma [38] showed hardness of training networks with no hidden neurons and a single output neuron with sigmoidal activation function. The latter paper contains an informative survey of the numerous hardness results that were known at that time.

Recently, the attention has been turned to networks using the so-called ReLU function $[x]_+ = \max\{0, x\}$ as activation function due to its extreme popularity in practice. Goel et al. [22] and Dey et al. [15] showed that it is even NP-hard to train a network with no hidden neurons and a single output neuron using the ReLU activation function. For hardness on other simple architectures using the ReLU activation function, see [11, 10, 6]. In order to prevent overfitting, we may stop training early, although the costs could still be reduced further. In the context of training neural networks, overfitting can be regarded as a secondary problem, as the problem only emerges after we have been

able to train the network on the data at all. Thus, none of the complexity theory papers on this subject address overfitting.

Besides training neural networks, there exist other NP-hard problems related to neural networks, e.g., continual learning [26]. Some of these training problems are not only NP-hard, but also contained in NP, implying that they are NP-complete. For instance, consider a fully-connected network with one hidden layer of neurons and one output neuron, all using the ReLU activation function. Here, it is not hard to see that the problem of deciding if total cost $\delta = 0$ can be achieved is in NP. We will show that the network does not need to be much more complicated before the training problem becomes $\exists\mathbb{R}$-complete, even when $\delta = 0$. Because it is easy to add inconsistent data points, one can also use our reduction to show $\exists\mathbb{R}$-hardness for any threshold $\delta > 0$.

## 2 Membership

In order to prove that NN-TRAINING is $\exists\mathbb{R}$-complete, we show that the problem is contained in the class $\exists\mathbb{R}$ and that it is $\exists\mathbb{R}$-hard (just as when proving NP-completeness of a problem). The first part is obtained by proving that NN-TRAINING can be reduced to ETR, while the latter is to present a reduction in the opposite direction. To see that NN-TRAINING $\in \exists\mathbb{R}$, we use a recent result by Erickson et al. [18]. Given an algorithmic problem, a *real verification algorithm* $\mathbf{A}$ has the following properties. For every yes-instance $I$, there exists a witness $w$ consisting of integers and real numbers. Furthermore, $\mathbf{A}(I, w)$ can be executed on the real RAM in polynomial time and the algorithm returns yes. On the other hand, for every no-instance $I$ and any witness $w$ the output $\mathbf{A}(I, w)$ is no. [18] showed that an algorithmic problem is in $\exists\mathbb{R}$ if and only if there exists a real verification algorithm. Note that this is very similar to how NP-membership is usually shown. The crucial difference is that a real verification algorithm accepts real numbers as input for the witness and works on the real RAM instead of the integer RAM.

It remains to describe a real verification algorithm for NN-TRAINING. As a witness, we simply describe all the weights of the network. The verification then computes the total costs of all the data points and checks if it is below the given threshold $\delta$. Clearly, this algorithm can be executed in polynomial time on the real RAM. Note that if the activation function is not piecewise algebraic, e.g., the sigmoid function $\varphi(x) = 1/1+e^{-x}$, it is not clear that we have $\exists\mathbb{R}$-membership as the function is not supported by the real RAM model of computation [18]. Specifically, Richardson's theorem states that computations involving analytic functions may be undecidable [33].

## 3 Reduction

In the following, we describe a reduction from the $\exists\mathbb{R}$-hard problem ETR-INV to NN-TRAINING. As a first step, we establish $\exists\mathbb{R}$-hardness of ETR-INV using previous work. As the next step, we define the intermediate problem RESTRICTED TRAINING. In the main part of the reduction, we describe how to encode variables, subtraction operations as well as inversion and addition constraints in RESTRICTED TRAINING. Finally, we present two modifications that enable the step from RESTRICTED TRAINING to NN-TRAINING.

### 3.1 Reduction to ETR-INV-EQ

In order to define the new algebraic problem ETR-INV-EQ, we recall the definition of ETR-INV. An ETR-INV *formula* $\Phi = \Phi(x_1, \ldots, x_n)$ is a conjunction $(\bigwedge_{i=1}^{m} C_i)$ of $m \geq 0$ constraints where each constraint $C_i$ with variables $x, y, z \in \{x_1, \ldots, x_n\}$ has one of the forms

$$x + y = z, \qquad x \cdot y = 1.$$

The first constraint is called an *addition* constraint and the second is an *inversion* constraint. An instance $I$ of the ETR-INV *problem* consists of an ETR-INV formula $\Phi$. The goal is to decide whether there are real numbers that satisfy all the constraints.

Abrahamsen et al. [5] established the following theorem. Note that their definition of ETR-INV formulas asks for a number of additional properties (e.g., restricting the variables to certain ranges) that we do not need for our purposes, so these can be omitted without affecting the correctness of the following result.

**Theorem A** ([5], Theorem 3). ETR-INV *is* $\exists\mathbb{R}$-*complete.*

For our purposes, we slightly extend their result and define the algorithmic problem ETR-INV-EQ in which each constraint has the form

$$x^{\pm 1} + y^{\pm 1} - z^{\pm 1} = 0.$$

We call a constraint of the above form a *combined constraint*, as it is possible to express both inversion and addition constraints using combined constraints. To see that ETR-INV-EQ is also $\exists\mathbb{R}$-complete, we show how to transform an instance of ETR-INV into an instance of ETR-INV-EQ. First, note that we can assume that every variable is contained in at least one addition constraint; otherwise, we can add a trivially satisfiable addition constraint, i.e., $x + y = y$ for a new variable $y$. Furthermore, consider the case that a variable $x$ of $\Phi$ appears in (at least) two inversion constraints, i.e., there exist two constraints of the form $x \cdot y_1 = 1$ and $x \cdot y_2 = 1$. This implies $y_1 = y_2$ and we can replace all occurrences of $y_2$ by $y_1$. Thus, we may assume that each variable appears in at most one inversion constraint. Now, if there is an inversion constraint $x \cdot y = 1$, we replace all occurrences of $y$ (in addition constraints) by $x^{-1}$. In this way, the inversion constraint becomes redundant and we can remove it. We are then left with a collection of combined constraints. This proves $\exists\mathbb{R}$-completeness of ETR-INV-EQ.

## 3.2 Reducing ETR-INV-EQ to RESTRICTED TRAINING

In the following, we reduce ETR-INV-EQ to NN-TRAINING. We start by defining the algorithmic problem RESTRICTED TRAINING which differs from NN-TRAINING in the following three properties:

- Some weights and biases can be predefined in the input.
- The output values of data points may contain a question mark symbol '?'.
- When computing the total cost, all output values with question marks are ignored.

Figure 1 displays an example of such an instance.

In the rest of the reduction, we will use the identity as activation function, and threshold $\delta = 0$. The reduction works for all honest cost functions.

**High-level Architecture.** The overall network consists of two layers, as depicted in Figure 1. Note that these layers are *not* fully connected. Some weights of the first layer will represent variables of the ETR-INV-EQ; other weights will be predefined by the input and some will have purely auxiliary purposes. Furthermore all biases are set to $0$.

### 3.2.1 The Gadgets

In the following, we describe the individual gadgets to represent variables, addition and inversion. Then we show how we combine these parts. Later, we modify the construction to remove preset weights, biases and question marks. At last, we will sketch how this reduction can be modified to work with ReLU functions as activation functions.

**Subtraction Gadget.** The subtraction gadget consists of five neurons, four edges, two prescribed weights and one data point. See Figure 2 (left) for an illustration of the network architecture of the

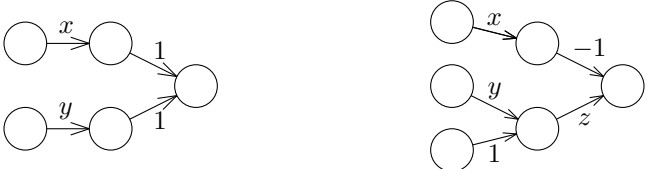

Figure 2: (left) The architecture of the subtraction gadget. The data point $d = (1, 1; 0)$ enforces that the constraint $x = -y$.
(right) The architecture of the inversion gadget. The data points $d_1 = (0, 1, 0; 1)$ and $d_2 = (1, 0, 1; 0)$ enforce $y \cdot z = 1$ and $x - z = 0$, respectively, implying that $x \cdot y = 1$.

subtraction gadget. The data point $d = (1, 1; 0)$ enforces that the constraint $x = -y$, as can be easily calculated.

**Inversion Gadget.** The purpose of the inversion gadget is to enforce that two variables are the inverse of one another. It consists of six vertices, five edges, two prescribed weights and two data points; for an illustration of the architecture see Figure 2 (right). A simple calculation shows that the data point $d_1 = (0, 1, 0; 1)$ enforces the constraint $y \cdot z = 1$, while the data point $d_2 = (1, 0, 1; 0)$ enforces the constraint $x - z = 0$. It follows that $x \cdot y = 1$.

**Variable Gadget.** For every variable, we build a gadget such that there exist four weights on the first layer with the values $x, -x, 1/x, -1/x$.

The variable-gadget is a combination of two subtraction gadgets and one inversion gadget. In total it has five input neurons, four middle neurons and two output neurons, see Figure 3 for an illustration of the architecture and the initial weights. We denote the input neurons by $s_1, \dots, s_5$. We denote the output neurons by $a, b$. The output neuron $a$ is drawn twice for clarity of the drawing. We have the data points $d_1 = (1, 1, 0, 0, 0; 0, ?)$, $d_2 = (0, 0, 1, 1, 0; 0, ?)$, $d_3 = (0, 0, 1, 0, 0; ?, 1)$, and $d_4 = (0, 1, 0, 0, 1; ?, 1)$.

The data point $d_1$ enforces $w = -x$. To see this note that the output neurons with a question mark symbol are irrelevant. Similarly, input neurons with 0-entries can be ignored. The remaining neurons form exactly the subtraction gadget. Analogously, we conclude that $y = -z$, using data point $d_2$. From the data point $d_3$, we infer that $y \cdot v = 1$. We infer $v = x$. using $d_4$, We summarize our observations in the following lemma.

**Lemma 3** (Variable-Gadget). *The variable gadget enforces the following constraints on the weights: $w = -x$, $y = 1/x$, and $z = -1/x$.*

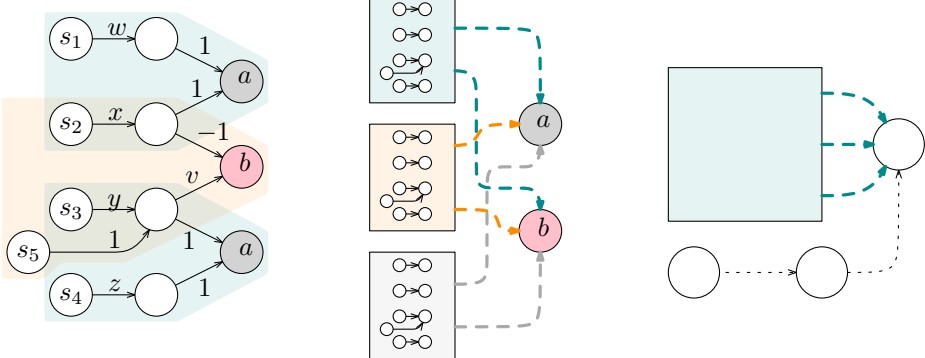

Figure 3: (left) The variable gadget. (middle) A schematic drawing of three combined variable gadgets. While the two output neurons of all gadgets are identified, all other neurons remain distinct. (right) For every '?' in a data point $d$, we add two more vertices and edges to the neural network.

**Combining Variable Gadgets.** Here, we describe how to combine $n$ variable gadgets. For the architecture, we identify the two output neurons of all gadgets; all other neurons remain distinct. Figure 3 gives a schematic drawing of the architecture for the case of $n = 3$ variables. Additionally, we construct $4n$ data points. For each variable, we construct four data points as described for the variable gadget; the additional input entries are set to $0$. In this way, we represent all $n$ variables of an ETR-INV-EQ formula. Furthermore, for each variable $x$, we have edges from the first to the second vertex layer, with the values $x, -x, 1/x$, and $-1/x$, see Lemma 3.

**Combined Constraint.** For the purpose of concreteness, we consider the combined constraint $C$ of an ETR-INV-EQ instance $w_1 + w_2 + w_3 = 0$, where each $w_i$ is either the value of some variable, its inverse, its negative or its negative inverse. Then, by construction, there exists a weight in the combined variable gadget for each $w_i$. Figure 4 depicts the network induced by the edges and the output vertex $a$; in particular, note that all the edges with weights $w_i$ are connected to $a$, as can also be checked in Figure 3.

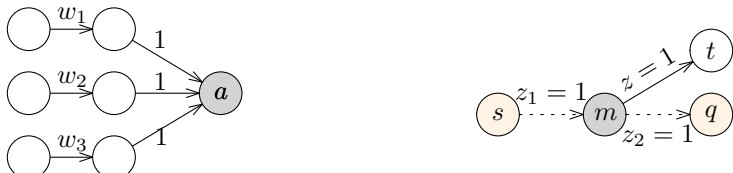

Figure 4: (left) There are three weights encoding $w_1$, $w_2$, and $w_3$. They are all connected to the output vertex $a$. (right) For each middle neuron $m$, we add an input neuron $s$ and the edges $sm$ and $mq$ with weights $z_1$ and $z_2$.

In order to represent the constraint $C$, we introduce a data point $d(C)$. It has input entry 1 exactly at the input neurons of $w_1$, $w_2$, and $w_3$; otherwise it is 0. Its output is defined by 0 for $a$ and '?' for $b$. Thus $d(C)$ enforces the combined constraint $C$. Note that enforcing the combined constraints does not require to alter the neural network architecture or to modify any of these weights.

### 3.2.2 Removing Fixed Weights

Next, we modify the construction such that we do not make use of predefined weights. To this end, we show how to enforce edge weights of $\pm 1$. Recall that, by construction, all predefined weights are either $+1$ or $-1$ and that the neural network architecture has precisely two layers. Furthermore, by construction, every middle neuron is incident to at least one edge with a weight that is specified by the input.

Globally, we add one additional output neuron $q$. For each middle neuron $m$, we perform the following steps individually: We add one more input neuron $s$ and insert the two edges $sm$ and $mq$. We will show later that the weights $z_1$ and $z_2$ on the two new edges can be assumed to be 1 as depicted in Figure 4. We modify all previously defined data points such that they have output '?' for output neuron $q$. They are further padded with zeros for all the new input neurons.

Furthermore, we add one data point $d(m)$ with input entry 1 for $s$ and 0 otherwise, and output entry 1 for $q$ and '?' otherwise. This data point ensures that neither $z_1$ nor $z_2$ are 0.

Next, we describe a simple observation. Consider a single middle neuron $m$, with $k$ incoming edges and $l$ outgoing edges. Let us denote some arbitrary input by $a = (a_1, \ldots, a_k)$, the weights on the first layer by $w = (w_1, \ldots, w_k)$, and the weights of the second layer by $\overline{w} = (\overline{w}_1, \ldots, \overline{w}_l)$. We consider all vectors to be columns. Then, the output vector for this input is given by $\langle a, w \rangle \cdot \overline{w}$, where $\langle \cdot, \cdot \rangle$ denotes the scalar product.

Let $\alpha \neq 0$ be some real number. Note that replacing $w$ by $w' = \alpha \cdot w$ and $\overline{w}$ by $\overline{w}' = 1/\alpha \cdot \overline{w}$ does not change the output.

**Observation 4.** *Scaling the weights of incoming edges of a middle neuron by $\alpha \neq 0$ and the weights of outgoing edges by $\alpha^{-1}$ does not change the neural network behavior.*

This observation can be used to assume that some non-zero weight equals 1 because we can freely choose some $\alpha \neq 0$ and multiply all weights as described above without changing the output. In particular, for the middle neuron $m$, we may assume that $z_1 = 1$, see Figure 4. Here, we crucially use the fact that $z_1$ is not zero. This standard technique is often referred to as *normalization*. Moreover, by the data point $d(m)$, we can also infer that the weight $z_2$ equals 1, as we know $z_1 \cdot z_2 = 1$.

With the help of the edge weights $z_1 = z_2 = 1$, we are able to set more weights to $\pm 1$. Let $z$ denote the weight of some other edge $e$ incident to $m$ that we wish to fix to the value $+1$ (the case of $-1$ is analogous). We describe the case that for some output neuron $t$ the edge $e = mt$ is an outgoing edge of $m$; the case of an incoming edge is analogous. See Figure 4 for an illustration. We add a new data point $d$, with output entries being 1 for $t$ and '?' otherwise and input entries being 1 for $s$ and 0 otherwise. Given that $z_1 = 1$, the data point $d$ implies that $z = 1$ as well.

### 3.2.3 Free Biases

In this section, we modify the input such that we do not make use of the biases being set to zero. First note that a function $f$ representing a certain neural network, might be represented by several weights

and biases. To be specific, let $b \in \mathbb{R}$ the bias of a fixed middle neuron $m$. Denote by $z_1, \ldots, z_k$ the weights of the outgoing edges of $m$ and $b_1, \ldots, b_k$ the biases of the corresponding neurons. We can replace these biases as follows: we replace $b$ by $b' = 0$ and $b_i$ by $b'_i = b_i + z_i \cdot b$. We observe that the new neural network represents exactly the same function $f$. Note that we used here explicitly the fact that the activation function is the identity. This may not be the case for other activation functions. From here on, we assume that all biases in the middle layer are set to zero.

It remains to ensure that the output neurons are zero as well. We add the additional data point $d = (\mathbf{0}, \mathbf{0})$ that is zero on all inputs and outputs. The value of the neural network on the input $\mathbf{0}$ is precisely the bias of all its output neurons. As our threshold is zero and the cost function is honest, we can conclude that the biases on all output neurons must be zero as well. We summarize this observations as follows.

**Observation 5.** *All biases can be assumed to be zero.*

### 3.2.4 Removing Question Marks

To complete the construction, it remains to show how to remove the question marks from the data points. We remove the question marks one after the other. For each data point $d$ and every contained symbol '?', we add an input neuron, a middle neuron, the edge between them and the edges from the input neuron to the output neuron containing the considered symbol '?' in $d$, see Figure 3 for a schematic illustration.

Due to the additional input entry for the added input neuron, we need to modify all data points slightly. In $d$, we set the additional entry to $1$; for all other data points, we set it to $0$. Moreover, we replace the considered symbol '?' in $d$ by the entry $0$.

We have to show that this modification does not change the feasibility of the neural network. Clearly, the output entries of $d$ with the question mark can now be freely adjusted using the two new edges. At the same time no other data point $d'$ can make use of the new edges as the value for the new input neurons equals to $0$.

This finishes the description of the reduction. Next, we show its correctness.

### 3.3 Correctness

Let $\Phi$ be an ETR-INV-EQ instance on $n$ variables. We construct an instance $I$ of NN-TRAINING as described above. First note that the construction is polynomial in $n$ in time and space complexity. To be precise, the size of the network is $O(n)$, the number of data points is $O(n)$, and each data point has a size in $O(n)$. Thus, the total space complexity is in $O(n^2)$. Because no part of the construction needs additional computation time, the time complexity is also in $O(n^2)$.

We show that $\Phi$ has a real solution $x^* \in \mathbb{R}^n$ if and only if there exists a set of weights for $I$ such that all input data are mapped to the correct output.

Suppose that there exists a solution $x^* \in \mathbb{R}^n$ satisfying all constraints of $\Phi$. We show that there are weights $w$ for $I$ that predict all outputs correctly for each data point. By construction, for every variable $x$, there exist edge weights $x, -x, 1/x, -1/x$. We set these weights to the value of $x$ given by $x^*$. Moreover, we prescribe all other weights as indented by the construction procedure; e.g., $\pm 1$ for the *prescribed* weights. By construction and the arguments above, all data points are predicted correctly by the neural network. Specifically, all the data points introduced for the combined constraints are correctly predicted, as $x^*$ satisfies $\Phi$.

For the reverse direction, we suppose that we are given weights $w$ for all the edges of the network in $I$. By Section 3.2.3, we can assume that all biases are zero without changing the function that is represents by the neural network. By Observation 4, we may normalize the weights without changing the behaviour of the neural network. Consequently, we can assume that all the weights are as prescribed for the RESTRICTED TRAINING problem. By Lemma 3, there exist edge weights that consistently encode the variables. Thus, we use the values of these weights to describe a real solution $x^*$ that satisfies $\Phi$. Due to the data points introduced for the combined constraints, we can conclude that all combined constraints of $\Phi$ are satisfied.

This finishes the proof of Theorem 2.

# 4  Conclusion

Training neural networks is undoubtedly a fundamental problem in machine learning. We present a clean and simple argument to show that NN-TRAINING is complete for the complexity class $\exists\mathbb{R}$. Compared to other prominent $\exists\mathbb{R}$-hardness proofs, such as [34] or [5], our proof is relatively accessible. Our findings illustrate the fundamental difficulty of training neural networks. At the same time, we give an indication on why neural networks can be a very powerful tool, since we prove neural networks to be more expressive than any learning method involving only discrete parameters or linear models: In practice, neural networks proved useful to solve problems that cannot be solved by combinatorial methods such as ILP solvers, SAT solvers, or linear programming, and our work gives a reason why (at least under the assumption that $NP \neq \exists\mathbb{R}$).

In our reduction, we carefully choose which edges should be part of our network to obtain an architecture that is particularly difficult to train. In practice, it is common to use a simpler *fully connected* network, where each neuron from one layer has an edge to each neuron of the next. It is thus an exciting and challenging open problem for future research to find out if training a fully connected neural network with one hidden layer of neurons is also $\exists\mathbb{R}$-complete, which we expect to be the case. We formulate the following conjecture.

**Conjecture 6.** NN-TRAINING *is $\exists\mathbb{R}$-complete, even for fully connected networks with one hidden layer and ReLU activation functions.*

Note that as mentioned in Section 1.4, this requires at least two output neurons, as the problem is otherwise in NP (when using ReLU or identity activation functions).

**Acknowledgments.**   We thank Frank Staals for many enjoyable and valuable discussions. We thank Thijs van Ommen for helpful comments on the write-up. We thank anonymous reviewers for their helpful comments and questions.

Mikkel Abrahamsen is part of Basic Algorithms Research Copenhagen (BARC), supported by the VILLUM Foundation grant 16582. Linda Kleist is supported by a postdoc fellowship of the German Academic Exchange Service (DAAD). Tillmann Miltzow is generously supported by the Netherlands Organisation for Scientific Research (NWO) under project no. 016.Veni.192.250.

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
