# OpenReview forum: "Training Neural Networks is ER-complete"
_NeurIPS.cc/2021/Conference — NeurIPS 2021 Poster_

### Official Review · Reviewer_hUB1 · 2021-07-03

**Rating:** 4
**Confidence:** 4

**Summary:**

This paper proves ETR-hardness of the problem of training a neural network to achieve a loss below a given threshold.
ETR complete problems are believed to be harder to solve than NP-complete problem, hence the results strengthen existing hardness results for training neural networks. The proof is via a reduction to the problem of deciding whether a polynomial equation with integer coefficients has a solution over the reals.

**Limitations And Societal Impact:**

Yes.

**Main Review:**

While the ETR hardness results are nice, I believe the conclusions made by the authors are misleading. The authors claim that  SAT-solving methods will not work well for training (and have not worked well in the past). This is not true: see for example the influential: "Reluplex: An Efficient SMT Solver for Verifying Deep Neural Networks". The suggestion that learning discrete structures such as grammars or Automates is easier than learning NN also does not fair well with the experience in ML, where training an NN is usually much easier than learning discrete models. The idea that NP-complete problems are "easy" and ETR problems are "hard" is strange. Both worst-case hardness results should imply intractability. The authors should spend more on discussing the worst case nature of their hardness results. Also the authors should be more careful in using the term local search (and claiming it will not work for training). After all, gradient based optimization is a local search  method.
Additionally, the paper proves hardness results for NN architectures that are rarely used in practice. Extending the results to more popular architectures (such as fully connected ones) would strengthen the results.   Another is to use ETR solvers to get algorithmic results in training neural networks.


**Time Spent Reviewing:**

8

---

> ### Author Response · Authors · 2021-08-06
> **Response**
>
> We thank the reviewer for the valuable feedback.
> In the following, we comment on some statements from the review.
>
> > The authors claim that SAT-solving methods will not work well for training (and have not worked well in the past).
>
> Thank you very much for this reference. However, the mentioned paper studies verifying neural networks rather than training them. While there exist special cases where it is possible to use other methods, such as SAT solvers, general neural networks are predominantly trained using gradient descent.
>
> > The idea that NP-complete problems are "easy" and ETR problems are "hard" is strange.
>
> We agree. By no means,  we wanted to imply that problems in NP are 'easy'.
> By containment of NP in ER, ER-hard problems are 'harder' than NP-hard problems.
> In particular, the tool set for problems in NP is more mature.
>
> > The authors should be more careful in using the term local search (and claiming it will not work for training). After all, gradient based optimization is a local search method.
>
> We are happy to clarify our used term more concretely.
> We refer to the established term local search in the Algorithm community.
> In particular, it refers to a method where, for some constant $k$, the solution is modified by exchanging $k$ elements with some other $k$ elements, see also
> https://en.wikipedia.org/wiki/Local_search_(optimization).

---

### Official Review · Reviewer_CGEk · 2021-07-11

**Rating:** 6
**Confidence:** 2

**Summary:**

The authors showed that the training of neural networks is $\exists\mathbb{R}$-complete, which implies the reason that techniques commonly used for solving large NP-complete problems do not work for neural network training.

**Limitations And Societal Impact:**

The authors adequately addressed the limitations.

**Main Review:**

Strengths:
+ The authors partially answered the question of why backpropagation works better for training neural networks than those methods commonly used for NP-complete problems.
+ The authors determined the algorithmic complexity of nn-training.

Weaknesses:
+ The main result assumes that the activation function $\phi(x)$ is the identity function, while the most commonly used activation is practice is ReLU. It would be interesting to show that Conjecture 6 is true.
+ As the authors already mentioned, it is interesting to see if training fully-connected neural networks can also fall into the category of $\exists\mathbb{R}$-complete.


Correctness:
+ The paper has no false claim to the best of my knowledge.

Clarity:
+ The overall structure of this paper is clear.

*After Rebuttal*

I have read through other reviews, and I agree that the notion of training in the paper is not as convincing as I originally thought it would be. However, I also agree with g6Te that there may be interesting and sound techniques established. I change my score to 6.



**Time Spent Reviewing:**

4 hours

---

### Official Review · Reviewer_GFLf · 2021-07-15

**Rating:** 5
**Confidence:** 4

**Summary:**

It is known that training neural networks is NP-hard, and this work makes a further attempt by studying the ER-hardness of the problem. The definition of 'neural network training' used in the proof of this paper is: do there exist weights such that the neural net maps every data input to corresponding output? This work proves that training neural networks is ER-complete, which is believed to be harder than NP.

**Limitations And Societal Impact:**

I appreciate that the authors provide some discussion on the limitation of the activation function. It would be much better if the architecture limitation and the threshold limitation are discussed in details.

**Main Review:**

**Originality**

It seems that this paper is the first to study the ER-hardness of training neural networks. So, good originality as far I see.

**Quality**

I find no errors in the proofs, and the overall quality is good in my opinion. However, I do find some arguments not well supported.

1. It is claimed in the abstract (line 1-4) that this paper proves that finding weights such that *the error is below a threshold* is ER-complete. However, theorem 2 requires *the error is 0*. Is it easy to extend theorem 2 to 'below a threshold'? If yes, then it should be discussed somewhere. If no, then the claim in the abstract is not appropriate.
2. In line 188, it is stated that some additional properties of the original ETR-INV can be ignored without affecting the correctness of the theorem A. Given that it is a theorem, I think it should be provided with proof.
3. There are three major differences between the regular neural networks and the neural networks used in this paper, i.e., (1) the neural network architectures considered in this paper are carefully constructed; (2) the activation function is always identity; (3) perfect mapping (zero error) is considered, rather than below a threshold. I appreciate the authors address the first limitation with a discussion about the ReLU activation. However, it is still interesting to see more discussion about the other two limitations. Rigorously speaking, to make *"Training Neural Networks is ER-complete"* such a title, it is expected that the above three limitations are fully discussed, if not solved.

**Clarity**

Overall the paper is well written, but I do have some complaints about how the proof is structured. For example, summarizing section 2 with a lemma may improve the clarity.

**Significance**

The hardness of training neural networks is fundamental and surely interesting. Proving it to be ER-complete is indeed a further step from NP-hard. However, the three limitations discussed above bring down the significance of this paper to some extent.

**Time Spent Reviewing:**

7

---

> ### Author Response · Authors · 2021-08-06
> **Response**
>
> We thank the reviewer for the valuable feedback.
> In the following, we comment on some statements/questions from the review.
>
> > Is it easy to extend theorem 2 to 'below a threshold'? If yes, then it should be discussed somewhere. If no, then the claim in the abstract is not appropriate.
>
> We prove that the problem is ER-hard even when the threshold is 0. Therefore, when the threshold is an arbitrary non-negative value given as part of the input, the problem is also ER-hard.
>
> > In line 188, it is stated that some additional properties of the original ETR-INV can be ignored without affecting the correctness of the theorem A. Given that it is a theorem, I think it should be provided with proof.
>
> Theorem A follows from the proof in reference [3]. We use a much weaker version of the theorem in [3]. Recently, there appeared a different paper with a similar statement that might serve as a better reference.
>
>
> >the activation function is always identity;
>
> When the special case where we only use the identity function is hard, the more general problem where each neuron has an activation function specified as part of the input is also hard.
> Note that many of the NP-hardness proofs for NN rely on the fact that the activation function is complicated.
> The fact that we use the identity shows that hardness does not come from the choice of a complicated activation function, as the identity is even simpler than the ReLU function.
>
>
>
> > There are three major differences between the regular neural networks and the neural networks used in this paper, i.e., (1) the neural network architectures considered in this paper are carefully constructed; (2) the activation function is always identity; (3) perfect mapping (zero error) is considered, rather than below a threshold. I appreciate the authors address the first limitation with a discussion about the ReLU activation. However, it is still interesting to see more discussion about the other two limitations. Rigorously speaking, to make "Training Neural Networks is ER-complete" such a title, it is expected that the above three limitations are fully discussed, if not solved.
>
> We are happy to extend our discussion and explain that limitation (3) does not exist and limitation (2) can also be interpreted as a strength. As mentioned in the paper, limitation (1) remains as an interesting open problem for future work.

---

### Official Review · Reviewer_g6Te · 2021-07-16

**Rating:** 9
**Confidence:** 4

**Summary:**

This paper proves that THE "Neural Network Training" problem is complete in a complexity class called ETR that is believed to be strictly larger than the class NP. Thus, it establishes a stronger hardness result than the well-known NP-Hardness for the problem, and shows that the problem is, in fact, probably not NP-Complete.

ETR stands for Existential Theory of the Reals and it can be thought of as a continuous analog of NP. The canonical complete problem for this class is similar to SAT by instead of boolean clauses we have algebraic constraints, and instead of a satisfying boolean assignment to the variables we are looking for an assignment of real values.

The result is proved by:
(1) Proving containment in ETR: this is the easy direction and follows from known methods for proving containment in ETR.
(2) Proving ETR-Hardness via a reduction from the canonical ETR problem. This is done by simple but clever gadgets that encode algebraic constraints with neural networks and data points.

This result is proved even for very restricted settings of the NN-Training problem, with one hidden layer and standard activation functions and so on.

**Main Review:**

I think this is a great submission giving a strong and fundamental theoretical result about the core problem of "NeurIPS" and ML. Moreover, as the authors explain, this theoretical result comes with some conceptual message: it sheds light on why neural networks are more expressive than other discrete structures, and on why SAT-solvers and other methods that are useful against NP-complete problems in practice, are not as effective against NN-Training (because it is not NP-complete!).

From a technical perspective the proof is not difficult but it requires some work and familiarity with recent progress on proving ETR completeness; a topic that is rather active these days in the theory community.

The paper is very well-written and provides clear motivations and explanations for a wide audience.

**Time Spent Reviewing:**

3

---

### Decision · Program_Chairs · 2021-09-27

**Decision:**

Accept (Poster)

**Comment:**

The paper shows that the problem of (**exact**) empirical risk minimization (ERM) in linear neural networks is ETR-Complete. The result is technically innovative, and interesting. Nevertheless it was felt by the reviewers that some of the conclusions made by the authors are misleading, and not very convincing.

While exact solving of the ERM objective is an interesting question, there is a crucial gap between ERM and training -- which normally refers to learning good parameters of the model that obtain small *excess* risk. Because of that, it is normally justified to find an $\textrm{OPT}+\epsilon$ solution for ERM. This is justified because $\epsilon$ error is incurred anyway due to statistical uncertainty. Moreover, in any reasonable model (where solution of the ERM can begin to lead to some generalization guarantees) finding an $\textrm{OPT}+\epsilon$ is regarded NP hard (by standard discretization/packing/covering and union bound arguments).

If the authors want to draw conclusions on the hardness of training from hardness of ERM solving, they would need to further justify this. In the final revision please tone down the last paragraphs in the introduction, and discuss the difference between *exact* ERM and training, as well as address other reviewers comments.